# Applying a Portable Backpack Lidar to Measure and Locate Trees in a Nature Forest Plot: Accuracy and Error Analyses

Yuyang Xie [1] , Tao Yang [2], Xiaofeng Wang [2], Xi Chen [2], Shuxin Pang [3] , Juan Hu [4], Anxian Wang [4], Ling Chen [4] and Zehao Shen [1,*]

1 Ministry of Education Key Laboratory for Earth Surface Processes, College of Urban and Environmental Sciences, Peking University, Beijing 100871, China; xie_yy@pku.edu.cn
2 School of Ecology and Environmental Science, Yunnan University, Kunming 650091, China; yt941112@ynu.edu.cn (T.Y.); wangxiaofeng@mail.ynu.edu.cn (X.W.); chenxi@mail.ynu.edu.cn (X.C.)
3 State Key Laboratory of Vegetation and Environmental Change, Institute of Botany, Chinese Academy of Sciences, Beijing 100093, China; pangshuxin@ibcas.ac.cn
4 Zhanyi Branch of Zhujiangyuan Provincial Nature Reserve Administration and Protection Bureau, Qujing 655331, China; zjybhqhj@163.com (J.H.); a2571181108@163.com (A.W.); cl13988994441@163.com (L.C.)
* Correspondence: shzh@urban.pku.edu.cn; Tel.: +86-010-62751179

**Abstract:** Accurate tree positioning and measurement of structural parameters are the basis of forest inventory and mapping, which are important for forest biomass calculation and community dynamics analyses. Portable backpack lidar that integrates the simultaneous localization and mapping (SLAM) technique with a global navigation satellite system receiver has greater flexibility for tree inventory than terrestrial laser scanning, but it has never been used to measure and map forest structure in a large area ($>10^1$ hectares) with high tree density. In the present study, we used the LiBackpack DG50 backpack lidar system to obtain the point cloud data of a 10 ha plot of subtropical evergreen broadleaved forest, and applied these data to quantify errors and related factors in the diameter at breast height (DBH) measurements and positioning for more than 1900 individual trees. We found an average error of 4.19 cm in the DBH measurements obtained by lidar, compared with manual field measurements. The incompleteness of the tree stem point clouds was the main factor that caused the DBH measurement errors, and the field DBH measurements and density of the point clouds also had significant impacts. The average tree positioning error was 4.64 m, and it was significantly affected by the distance and route length from the measured trees to the data acquisition start position, whereas it was affected little by the habitat complexity and characteristics of tree stems. The tree positioning measurement error led to increases in the mean value and variability of paired-tree distance error as the sample plot scale increased. We corrected the errors based on the estimates of predictive models. After correction, the DBH measurement error decreased by 31.3%, the tree positioning error decreased by 44.3%, and the paired-tree distance error decreased by 56.3%. As the sample plot scale increased, the accumulated paired-tree distance error stabilized gradually.

**Keywords:** backpack lidar; closed forest; SLAM; stem positioning; accuracy; paired-tree distance





## 1. Introduction

Constructing investigation plots to monitor forest dynamics is an important approach in plant community ecology research, and it also sets up a long-term platform for interdisciplinary biodiversity science [1]. These studies involved the inventory of individual trees in sampling plots, including determining the position and structural parameters of trees. However, ordinary forest inventory construction is time-consuming and laborious, especially in plots with complex terrain and areas larger than 10 ha [2]. Rangefinder, total station, and real-time kinematic systems have begun to be used for single tree positioning; the efficiency of these tools is low because the global navigation satellite system (GNSS)

signal and light transmission route are always blocked by the generally high density of canopy found in natural forests [3].

Lidar (light detection and ranging) systems can be used as a new approach to acquiring three-dimensional (3D) structural measurements of vegetation [4]. Ground-based terrestrial laser scanning (TLS) has been used widely to obtain structural parameters with high precision [5–7], but its fixed position limits its spatial flexibility. Therefore, TLS is inefficient for use in a large area with complex environments [8]. Mobile lidar systems, such as backpack lidar, can overcome this problem because they are portable, and the accuracy of tree structure measurement based on mobile lidar has also been confirmed [8–12]. However, most of the studies conducted to verify the measurement accuracy were performed in plantations with simple tree structures and orderly arrangements [8], or small-area natural coniferous forest plots with flat terrain [10], and only several dozen or hundreds of trees were selected for the validation set with little sheltering around; thus the quality of lidar point clouds was usually high and homogeneous. The high tree density and complex understory structure in natural forests in subtropical regions would lead to great differences in the quality of point clouds; thus, the suitability of backpack lidar for extracting natural large-scale forest structures needs to be further verified.

Mapping trees in forest sampling plots is the basis for exploring the structure and dynamics of forest communities [13–15], and for quantifying interspecific or intraspecific competition [16]. Thus, it is critical to obtain accurate position and structural measurements of trees. Integrating GNSS and lidar can provide accurate position information for point clouds and improve the efficiency and accuracy of automatic data registration [17,18]. However, the GNSS signal is usually shielded by the dense forest canopy. Therefore, it is particularly important to evaluate the potential of the core technique of automatic point cloud registration comprising simultaneous localization and mapping (SLAM) for mobile lidar application in such a complex environment. SLAM is a technique that can instantly build and update the surrounding environment's map or 3D information by tracking the sensor position; thus, it can provide an alternative environmental navigation and positioning method without a GNSS signal [9,19].

The SLAM system first extracts some features of scanned objects in the initially obtained point cloud data [20] and during movement; the system then calculates the rotation and translation matrices based on the features in the current and previous time step, before registering the point cloud and inertial measurement unit measurements, and updating the current trajectory and position information. All of the point data collected from the beginning of the data collection process, until the current time, are then updated to generate 3D images [3]. The accuracy of this technique has been verified in many mapping studies [21]. However, most of the research objects were architectural or indoor facades, and only a few studies have assessed the suitability of the technique for complex forests [3]. In particular, Qian et al. [17] quantified the difference in the tree positioning accuracy between sparse and dense forests using vehicle-mounted mobile lidar equipment. Chudá et al. [22] compared the tree positioning accuracy with different types of portable lidar. However, these studies did not quantify the effects of factors that caused positioning errors, while the former study only investigated about 1 ha of dense forest and the latter only targeted dozens of trees. Therefore, it is still necessary to determine whether this technique is suitable for inventory and tree mapping in large-scale forests.

The diameter at breast height (DBH) and position of each tree are basic measurements of forest structure—the distance between any two trees (paired-tree distance) is essential to calculate forest density—and these metrics are also necessary for constructing forest competition indexes [23–25]. In the present study, we used a backpack lidar system (LiBackpack) equipped with double Velodyne Puck VLP-16 laser sensors to collect complete point cloud data for a dense natural subtropical mixed forest of 10 ha in area, and aimed to determine: (1) the errors in DBH measurements and tree positioning obtained using the backpack lidar; (2) how different factors might affect the errors; and (3) whether the accuracy of the measurements could be optimized based on the relationships between

the measurement error and related factors. Our findings demonstrate the feasibility of obtaining backpack lidar measurements in a complex forest environment, and also provide suggestions for further improving and optimizing the accuracy of data so as to enhance the efficiency of forest inventory and monitoring.

## 2. Materials and Methods

### 2.1. Study Area

The study site comprised an area of 10 ha (200 × 500 m$^2$) in a semi-humid evergreen broad-leaved forest as the forest monitoring plot (103.9°E, 25.9°N; 2300 m a.s.l.) located at Maxiong Mountain, Pearl River Source Nature Reserve, Yunnan Province, China. This reserve is located on the Nanpan River, which is one of the sources of the Pearl River, and it has high value for biodiversity maintenance, water storage, soil conservation, climate regulation, and tourism. This forest was destroyed by a wildfire 30 years ago and then underwent restoration without much human disturbance. The establishment of a plot will provide an important platform for the observation of natural community succession. The tree density in the large sample plot was about 8600 trees/ha. The dominant species in the tree layer was *Castanopsis fargesii*, which accounted for more than 90% of all trees, and the dominant shrubs were *Rhododendron irroratum* and *Rhododendron delavayi*.

### 2.2. Field Measurement

The large forest plot (Figure 1) was artificially divided into 10 × 25 small survey grids, each with an area of 20 × 20 m$^2$. In this process, 286 grid corners were accurately located by using the total station Z TS-420R (Hi-Target, Guangzhou, China). Six to ten trees were selected in different directions around each grid corner, within a distance of 10 m, and accurately located ($X_{field}$, $Y_{field}$) using tape and a compass based on the precise coordinates of the nearest grid corners. At the same time, we used a diameter ruler to measure the DBH ($DBH_{field}$) at a height of 1.3 m for each selected tree.

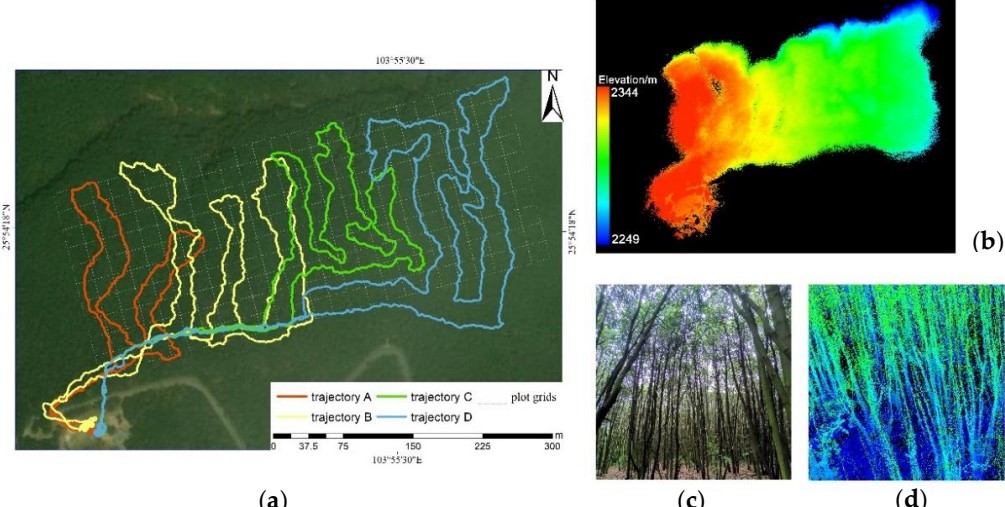

**Figure 1.** Situation in the forest plot and lidar data acquisition results. (**a**) Grids in the large forest plot and trajectories of multiple lidar scans. (**b**) Lidar point cloud data displayed by height. (**c**) Real scene in the forest plot and (**d**) the corresponding point cloud.

### 2.3. Lidar Data Collection

We used the LiBackpack DG50 backpack lidar system (Digital Green Valley Technology Co., Ltd., Beijing, China). This system is equipped with two Velodyne Puck VLP-16 lidar sensors that rotate in the horizontal and vertical directions, and high-precision GNSS equipment. The vertical field angle was 30°, and the measurement range was within 100 m, where 600,000 laser pulses were emitted per second. In the data acquisition process, a measurer

walked along a zigzag route in the forest with the backpack lidar equipment on his/her back. The start and end of the walking route were in the same open area, with a stable GNSS signal outside the sample plot, in order to ensure that part of the point cloud had geographic location information. The original point cloud data and GNSS location records were then imported into LiFuser BP software (https://www.greenvalleyintl.com/LiFuser/ accessed on 1 June 2021) for matching to obtain point cloud data containing geographic location information and the trajectories for data acquisition processing (Figure 1b). Due to the limited endurance of the equipment, we split the data acquisition into four parts (Figure 1a).

### 2.4. Data Processing

### 2.4.1. Preprocessing of Lidar Point Clouds

We extracted ground points from point clouds using an improved progressive triangulated irregular network densification algorithm [26] and subtracted the elevations of ground points from the elevations of the nearest non-ground points. Thus, the locations of all trees were transformed onto a horizontal plane. We gridded the ground points as a digital elevation model with a resolution of 0.5 m using an inverse distance-weighted interpolation method. These processes were implemented in LiDAR 360 V5.0 software (https://www.greenvalleyintl.com/LiDAR360/ accessed on 1 June 2021).

### 2.4.2. DBH Fitting and Trees Positioning

According to the measured positions and relative distributions of trees near the plot corners, we searched the preprocessed point cloud to find 3D models corresponding to the trees that we measured in the field. We applied the cylindrical fitting method based on 3D ordinary least squares to obtain the lidar DBH values ($DBH_{lidar}$) and stem coordinates ($X_{lidar}$,$Y_{lidar}$) for the trees. As described by Xie et al. [8], the height of the stem point cloud slices used in the algorithm was 30 cm (between 1.15–1.45 m from the ground) to optimize the fitting accuracy. These processes were implemented in LiDAR 360 V5.0 software. We calculated $\Delta DBH$ ($DBH_{lidar}$-$DBH_{field}$) as the DBH estimation error for lidar, and calculated $\Delta X$ ($X_{lidar}$-$X_{field}$) and $\Delta Y$ ($Y_{lidar}$-$Y_{field}$) as the lidar estimation errors for trees positioned in the east and south, respectively. We calculated $\Delta P$ as the square root of the sum of $\Delta X2$ and $\Delta Y2$ to represent the lidar tree positioning error. $\Delta X$, $\Delta Y$, and $\Delta P$ are shown in Figure 2a.

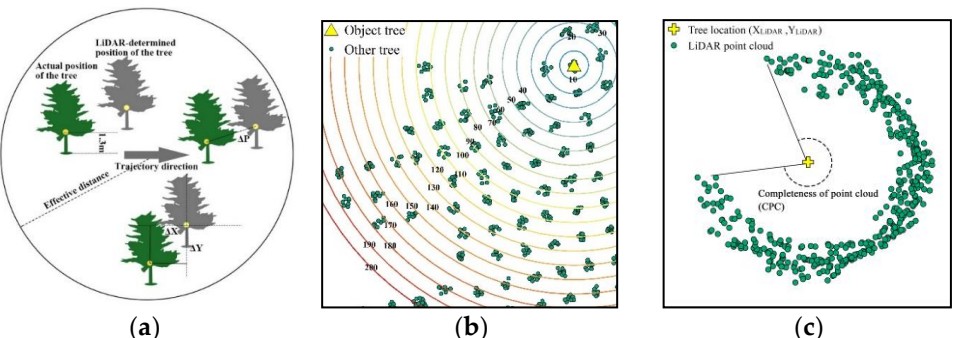

(**a**) (**b**) (**c**)

**Figure 2.** Schematic illustrations of indicators considered in this study. (**a**) Relative position error $\Delta P$ and deviation in the east and north offsets: $\Delta X$, $\Delta Y$. (**b**) Distance interval from any tree to other trees, i.e., search radius for calculating the relative distance error. The number in the figure is the radius of the search range with the target tree as the center (m). (**c**) Completeness of the stem point cloud (CPC, °).

### 2.4.3. Paired-Tree Distance

The growth of trees in forests is regulated by competition [27,28] and the paired-tree distance is often used as an indicator of competition intensity [29]. In order to explore the measurement error in the paired-tree distance and the scale effect, we set the locations of trees measured in the field ($X_{field}$, $Y_{field}$) as centers to establish a series of buffers (Figure 2b), which we defined as the search range for calculating paired-tree distance, and calculated the distances measured by lidar ($D_{lidar}$) and in the field ($D_{field}$) from each tree to all other trees

within each search range. We defined the measurement error in the paired-tree distance as $\Delta D = D_{lidar} - D_{field}$, and calculated the mean and standard deviation of $\Delta D$ for different search radius ranges in further analyses.

### 2.4.4. Extraction of Influencing Factors

In order to explore the factors that affected the accuracy of DBH measurements, we calculated the completeness of the stem point cloud (CPC), point cloud density (PCD), point cloud intensity (PCI), tree distance to trajectory (DT), terrain slope, slope aspect change rate (SOA), terrain roughness, and $DBH_{field}$ as independent variables. Due to sheltering by other trees, it is difficult to ensure that laser echoes are generated in all directions from a tree stem, so the shape of the stem point cloud is usually an incomplete cylinder. Considering that the incompleteness of the cylinder may have affected the cylinder fitting algorithm, we defined IPC (Figure 2c) as one radian on the cylindrical surface of the stem point cloud with the tree position as the axis. The CPC was calculated as follows. First, the 30 cm stem point cloud slice used for DBH fitting on the horizontal plane was orthogonally projected to obtain point cloud rings before calculating the azimuth radian value (°) based on the tree position for all points in the rings, and then sorted the azimuth radian values for all points corresponding to each tree from small to large, including 0° and 360°. We defined outliers in a sequence as those greater than the upper quartile plus 1.5 times the interquartile distance using the function boxplot.stats() in R (V4.1.2). We treated the sum of three maximum outliers of the first-order difference sequence of the azimuth radian sequence as the radian missing from the projection point ring described above. Finally, we subtracted the missing radian from 360° to obtain the CPC.

Previous studies have shown that PCD and $DBH_{field}$ can significantly affect the DBH measurements in plantations [8,30]; thus, they were also considered in the present study. PCD was calculated as the number of points on the unit horizontal projection [8]. PCI is the laser echo intensity on stems collected by the receiving device, which is related to the surface material, roughness, incident angle direction of the target, emission energy of the instrument, laser wavelength, and other attributes that were difficult to measure. DT is the average Euclidean distance from a single tree position to the trajectory. The terrain slope, SOA, and roughness reflecting the terrain complexity were calculated directly from the digital elevation model data and extracted to the position of each tree.

To explore the factors that affected the positioning accuracy of trees, in addition to the variables of the tree attributes specified above, we considered that the complex environment would interfere with the movement of the experimenter carrying the lidar, thereby influencing the scanning time, incident angle, as well as the performance of the SLAM algorithm. The complex environment provided more reference points for automatic data registration [17], but the greater numbers of branches and leaves also increased the variability of the incident surface material for the laser. We calculated the number of trees from the trajectory to each tree, as well as the average intensity, density, and total number of stem laser points, terrain slope, SOA, and roughness to reflect the complexity of the environment around the trees. The number of trees between the trajectory and the target tree was calculated using the bottom-up single tree segmentation algorithm [31] combined with visual interpretation.

In addition, the data accuracy of the point cloud may have been affected after losing the GNSS signal, because we could only obtain the starting position before entering the sample plot, and the GNSS signal was fixed in the whole process. Thus, we calculated the Euclidian distance between each tree and the starting position. Since the systematic error of the SLAM algorithm would accumulate during the whole process [21], we calculated the cumulative route length from the first laser echo generated on each tree to the starting position based on the accurate GPS time for every laser point and trajectory. We also calculated the standard deviation for the GPS time of laser points on each stem to represent the number of times the tree was scanned.

Among the indicators described above, the CPC and route length to the starting point were calculated using R (V4.1.2), PCI was the original value produced by LiFuser BP software, the number of trees was calculated with LiDAR 360 software, and the other indicators were calculated using ArcGIS v10.3.

### 2.5. Data Analysis

To estimate the contributions of different factors to the lidar measurement errors, we used random forest models to explain $\Delta DBH$ and $\Delta P$. We applied the bootstrap sampling method of 100 iterations, with 80% random samples drawn each time to build the random forest models, and the pseudo-R-squared value was used to measure the variation explained by the model. We used the increase in the node purity to measure the impact of each variable on the heterogeneity of the observed values at each node in the classification tree, and the weight contributed by each variable was expressed as the proportion for the variable relative to the total increase in the node purity value. The random forest model was constructed using the "randomForest" package in R (V4.1.2).

To measure the marginal responses of the measurement errors to different explanatory variables, we first conducted ten-fold cross-validation and used the principle of optimal subset regression to determine the optimum number of variables in the random forest model for selecting the optimal combination of variables. The "partialplot" function in the "randomForest" R package was then used to produce partial correlation diagrams for $\Delta DBH$ and $\Delta P$ with their influencing variables.

To describe the trend in each marginal response, according to the partial correlation between $\Delta DBH$ or $\Delta P$ and the dependent variables, we fitted univariate linear, quadratic, logarithmic, exponential, and logistic models, and then identified the best model to explain the variations in $\Delta DBH$ or $\Delta P$ with the dependent variables based on $R^2$. If the trend exhibited obvious segmentation characteristics, the "segmented" function in R (V4.1.2) was used to determine the segmentation position and to perform segmentation fitting.

Using the optimal univariate models determined above, we constructed a multivariate generalized linear model (GLM) with $\Delta DBH$, $\Delta X$, and $\Delta Y$ as dependent variables. According to $\Delta DBH = DBH_{lidar} - DBH_{field}$, $\Delta X = X_{lidar} - X_{field}$, and $\Delta Y = Y_{lidar} - Y_{field}$, the linear models were transformed into the form of the field-measured value ~ LiDAR-estimated value to obtain the transformation relationship between the two measurements.

Next, we corrected $DBH_{lidar}$, $X_{lidar}$, and $Y_{lidar}$ with the conversion model to obtain the real values and determined the corrected values of $\Delta P$ and $\Delta D$. We used one-way analysis of variance to test the differences between the lidar measurement errors before and after correction; then, we used locally weighted regression to explore the before- and after-correction difference of the trends in the mean and standard deviation of the $\Delta D$ values as the search range increased.

## 3. Result

### 3.1. DBH Measurement Error and Correction

The random forest model explained $30.40 \pm 0.015\%$ of the variation in $\Delta DBH$. Ten-fold cross-validation showed that the first three variables should be selected for the best model. IPC explained the most variation, where it accounted for $26.28 \pm 0.93\%$ of the total variation, and $DBH_{field}$ and PCD explained $17.96 \pm 0.63\%$ and $9.00 \pm 0.35\%$, respectively (Figure 3a). The DBH lidar measurement error ($\Delta DBH$) was $2.16 \pm 5.61$ cm, and the absolute error ($|\Delta DBH|$) was $4.19 \pm 4.30$ cm (Figure 4b).

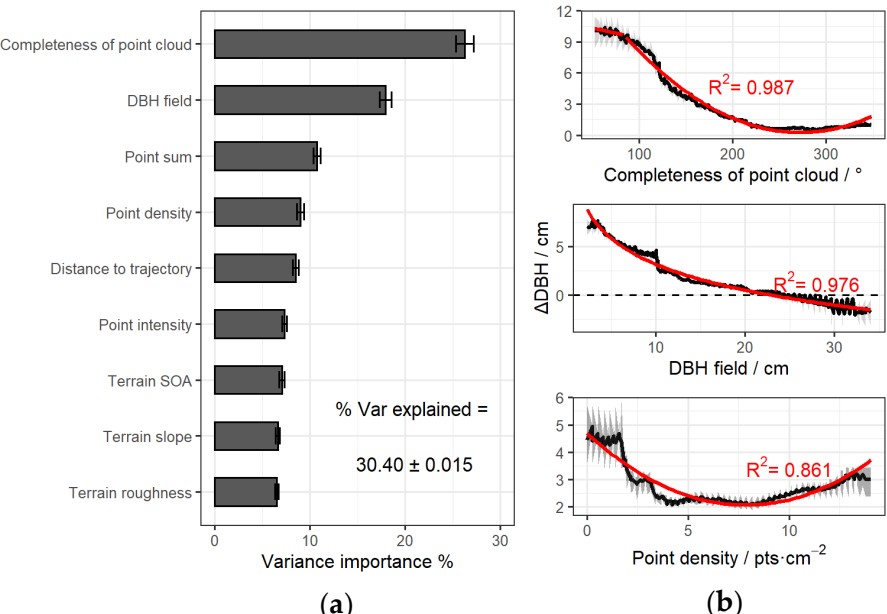

**Figure 3.** Regression results obtained with the random forest model for the DBH measurement error ΔDBH. (**a**) Relative importance of variables, where the error line is the standard deviation of the results obtained using the random forest model constructed by randomly drawing 80% of the samples 100 times. (**b**) Marginal response of the three most important variables, where the black lines are the average marginal response values obtained using the random forest model constructed by randomly drawing 80% of the samples 100 times. The upper and lower bounds of the gray shading are the maximum and minimum values, respectively, and the red curve is the best curve obtained by model screening based on $R^2$.

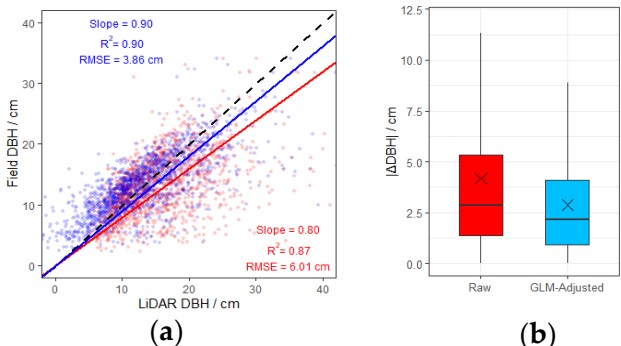

**Figure 4.** Relationship between the lidar-estimated and field-measured DBH values, and the estimation absolute error before and after correction based on the transformation model. (**a**) Cross-origin linear fitting of DBH lidar estimated and field measured values. The red points and line are the fitting results before correction. The blue points and line are the fitting results after correction. The black dotted line represents y = x. (**b**) Absolute errors (|ΔDBH|) of lidar measurements before and after correction. In the figure, × denotes the mean values. The differences are significant according to analysis of variance ($p < 0.05$).

According to the $R^2$ values, the quadratic model was the best in explaining the marginal response of ΔDBH to CPC ($R^2 = 0.978$; Figure 3b, Table 1). As the CPC increased, ΔDBH decreased and gradually stabilized. ΔDBH was minimized when CPC reached about 250°, being stable as CPC increased further.

**Table 1.** Curve fitting for the marginal response of the DBH measurement error (ΔDBH) obtained with the random forest model. The table shows the F value and $R^2$ value for the model.

| Variable | | Linear | Logarithmic | Quadratic | Exponential | Logistic |
|---|---|---|---|---|---|---|
| Completeness of point cloud (CPC) | F | 1274.358 | 3577.328 | 7804.132 | 2182.511 | 2033.291 |
| | $R^2$ | 0.786 | 0.912 | **0.978** | 0.863 | 0.854 |
| DBH field ($\text{DBH}_{\text{field}}$) | F | 3572.559 | 16,223.59 | 6269.308 | — | — |
| | $R^2$ | 0.901 | **0.976** | 0.97 | — | — |
| Point density (PD) | F | 44.255 | 204.248 | 598.36 | 32.151 | 45.977 |
| | $R^2$ | 0.183 | 0.508 | **0.861** | 0.14 | 0.188 |

The logarithmic model was optimal for explaining the marginal response of ΔDBH to $\text{DBH}_{\text{field}}$ ($R^2$ = 0.976; Figure 3b, Table 1). As $\text{DBH}_{\text{field}}$ increased, ΔDBH decreased and reached zero when $\text{DBH}_{\text{field}}$ was about 23 cm. Moreover, ΔDBH was < 0 for trees with $\text{DBH}_{\text{field}}$ > 23 cm, i.e., $\text{DBH}_{\text{lidar}}$ was less than $\text{DBH}_{\text{field}}$.

The quadratic model was selected for explaining the marginal response of ΔDBH to PCD ($R^2$ = 0.861; Figure 3b, Table 1). ΔDBH decreased rapidly initially and then increased as PCD increased. The minimum ΔDBH corresponded to the PCD of 8 points/$\text{cm}^2$.

Based on the relationship identified, we constructed a predictive model for ΔDBH (Equation (1)), where the $R^2$ value was 0.323, which was similar to the total variation explained by the random forest model. According to ΔDBH = $\text{DBH}_{\text{lidar}}$ − $\text{DBH}_{\text{field}}$, we set $\text{DBH}_{\text{field}}$ as $\text{DBH}_{\text{real}}$ and transformed Equation (1) into Equation (2) as the model for converting lidar measurements into real values, as follows.

$$\Delta\text{DBH} = -0.36\text{DBH}_{\text{field}} - 0.125\text{IPC} + 0.0002\text{CPC}^2 - 0.34\text{PD} + 0.24\text{PD}^2 + \mu \qquad (1)$$

$$\text{DBH}_{\text{real}} = 1.56\text{DBH}_{\text{Lidar}} + 0.195\text{IPC} - 0.0003\text{CPC}^2 + 0.53\text{PD} - 0.038\text{PD}^2 + \mu \qquad (2)$$

$\text{DBH}_{\text{field}}$ was also an explanatory variable, so, in order to ensure that $\text{DBH}_{\text{lidar}}$ had a unique value, the linear form of $\text{DBH}_{\text{field}}$ was considered instead of the logarithmic form. In the equations above, $\mu$ is the unexplained part, and the corrected $\text{DBH}_{\text{lidar}}$ is the part on the right-hand-side of the equation, excluding $\mu$. After correction, the linear fit ($R^2$) between $\text{DBH}_{\text{lidar}}$ and $\text{DBH}_{\text{field}}$ was improved, and the fitted cross-origin line was closer to y = x (slope = 0.9, slope before correction = 0.87; Figure 4a). The root mean square error decreased from 6.01 cm to 3.86 cm (Figure 4a), and the absolute estimation error (|ΔDBH|) decreased from 4.19 ± 4.30 cm to 2.88 ± 2.56 cm (Figure 4b), i.e., a decrease of 31.3%.

### 3.2. Tree Positioning Error and Correction

Figure 5 shows the distribution of the lidar tree positioning error (ΔP), where the total ΔP was 4.64 ± 2.50 m. The random forest model explained 83.66% ± 0.01% of the variation in ΔP, and the ten-fold cross-validation selected the first two variables for the best model. The distance to the start (DS) explained most of the variation (48.81 ± 1.03%) in ΔP, and the route length from the start (RS) explained 19.93 ± 0.66% (Figure 6a).

The model screening results based on the $R^2$ values are shown in Table 2. The quadratic model was selected for the marginal response of ΔP to DS ($R^2$ = 0.936), and the goodness-of-fit increased further after piecewise fitting with DS = 280 m as the boundary ($R^2$ = 0.987; Figure 6b). The variation of ΔP was only about 0.5 m before DS reached 280 m as a turning point. ΔP then increased significantly with the increase of DS. The logarithmic model was selected for the response of ΔP to RS ($R^2$ = 0.750; Figure 6c). ΔP increased with RS, but the rate fluctuated and slowed down before finally stabilizing.

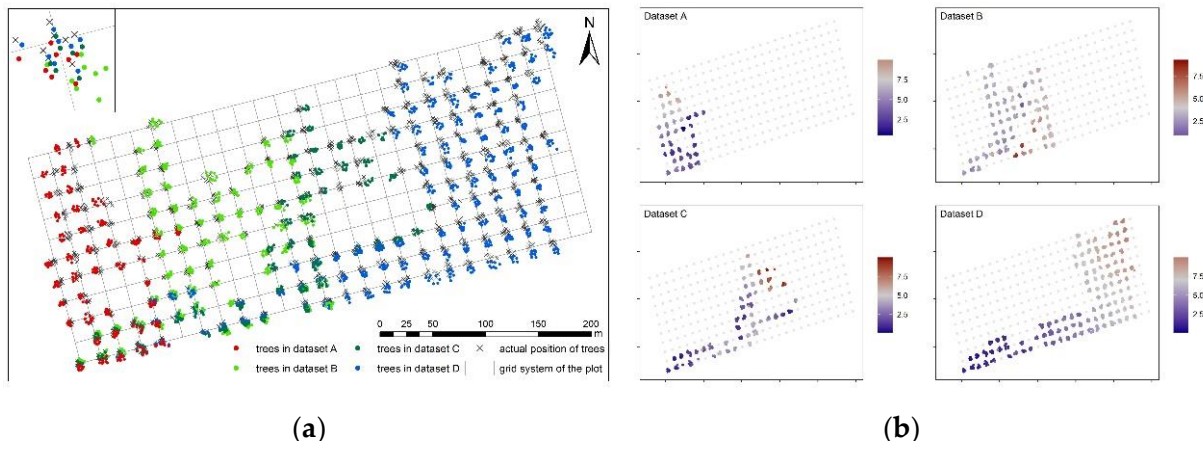

**(a)**                    **(b)**

**Figure 5.** Tree positioning and tree position error in lidar measurements. (**a**) Tree positioning results for each lidar point cloud data set and tree positions measured in the field. (**b**) Spatial distribution of lidar tree position error (ΔP, m). The color of the point indicates the error value. Datasets A, B, C, and D correspond to trajectories A, B, C, and D in Figure 1.

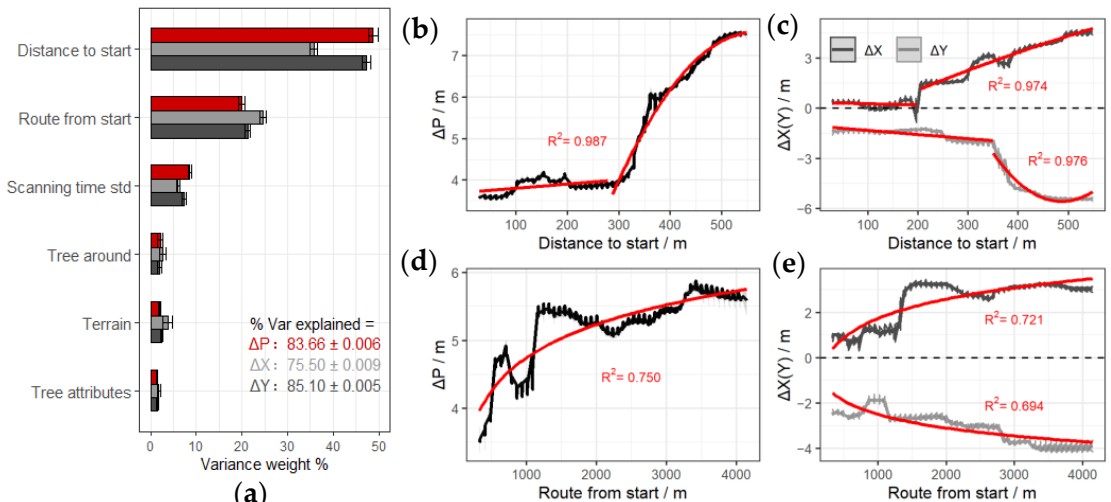

**Figure 6.** Results of random forest model for ΔP, ΔX, and ΔY. (**a**) Relative importance of variables. The error lines are the standard deviation of results constructed by randomly drawing 80% of the samples 100 times. Factors with minimal contributions were combined. (**b–e**) Marginal responses of the two most important variables. The black lines are the average marginal response values constructed by randomly drawing 80% of the samples 100 times. The upper and lower bounds of the gray shading are the maximum and minimum values, respectively, and the red curve is the best curve obtained by model screening based on $R^2$.

**Table 2.** Curve fitting for the marginal response of the ΔP values to the primary predictive variables. The table shows the F value and $R^2$ value for the models.

| Variable | | Linear | Logarithmic | Quadratic | Exponential | Logistic |
|---|---|---|---|---|---|---|
| Distance to start (DS) | F | 2202.59 | 658.00 | 3244.53 | 2380.83 | 1767.65 |
| | $R^2$ | 0.83 | 0.60 | **0.94** | 0.84 | 0.80 |
| Route from start (RS) | F | 563.65 | 1044.69 | 426.50 | 495.14 | 788.05 |
| | $R^2$ | 0.62 | **0.75** | 0.71 | 0.59 | 0.69 |

The random forest model explained 75.50% ± 0.01% and 85.10% ± 0.01% of the variation in the ΔX and ΔY axes, respectively, and the order of importance of the variables

was the same as that in ΔP model (Figure 6a). The trends in the marginal responses of ΔX and ΔY to DS and RS were similar to that for ΔP (Figure 6d,e); thus, they could be fitted with models of the same form. The inflection points of ΔX and ΔY corresponded to DS values of 200 m and 350 m, respectively (Figure 6d).

We constructed predictive models for ΔX and ΔY (Equations (3) and (4)) based on their relationships with *DS* and *RS*.

$$\Delta X = 3.73 \times e^{-5}DS^2 - 8.58 \times e^{-3}DS + 1.20 \ln RS + \mu \tag{3}$$

$$\Delta Y = -4.40 \times e^{-5}DS^2 + 1.17 \times e^{-2}DS - 2.90 \ln RS + \mu \tag{4}$$

In Equations (3) and (4), $\mu$ is the unexplained variance. The $R^2$ values for the GLMs were 0.443 and 0.528, respectively, which were clearly lower than those for the random forest models. According to $\Delta X = X_{lidar} - X_{field}$ and $\Delta Y = Y_{lidar} - Y_{field}$, and by setting ($X_{field}$, $Y_{field}$) as the real location ($X_{real}$, $Y_{real}$), we transformed the models to link values between the lidar-measured and field-measured positions (Equations (5) and (6)):

$$X_{real} = X_{LiDAR} - 3.73 \times e^{-5}DS^2 + 8.58 \times e^{-3}DS - 1.20 \ln RS + \mu \tag{5}$$

$$Y_{real} = Y_{LiDAR} + 4.40 \times e^{-5}DS^2 - 1.17 \times e^{-2}DS + 2.90 \ln RS + \mu \tag{6}$$

After correction, ΔX and ΔY decreased significantly, and ΔP decreased to $2.58 \pm 1.90$ m, 44.3% lower than the original measurement.

### 3.3. Measurement Error and Correction of Relative Distance between Trees

After tree position correction described above, the accuracy of the paired-tree distance improved significantly ($p < 0.05$; Figure 7a). The lidar measurement error of the mean paired-tree distance (ΔD) was $1.74 \pm 3.45$ m based on the raw point cloud data, and $0.76 \pm 3.46$ m after correction with model-adjusted data, i.e., ΔD decreased by 56.3%. When two trees were from the point cloud data of the same scanning, ΔD was $0.48 \pm 2.23$ m and $0.01 \pm 2.52$ m after correction. When two trees were from different point cloud data and scanned twice, ΔD was $2.27 \pm 3.72$ m and $1.08 \pm 3.74$ m after correction.

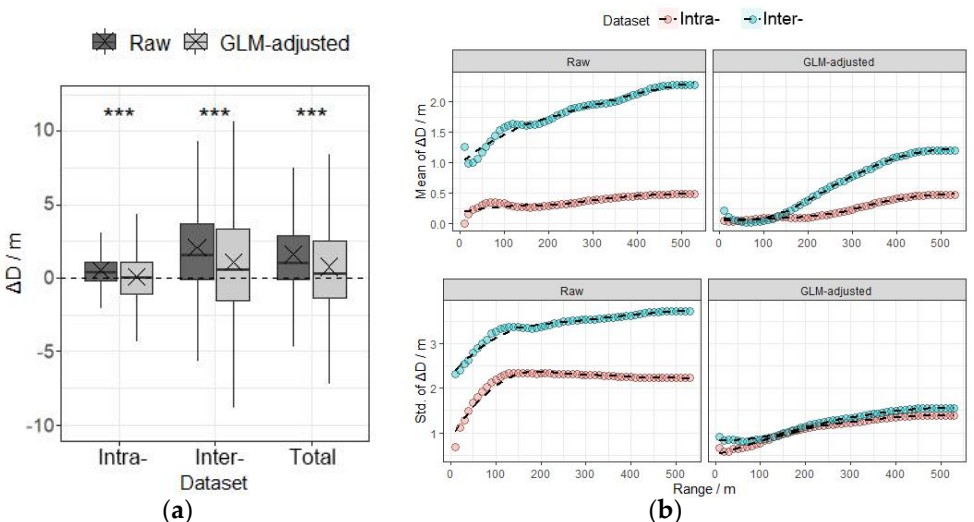

**Figure 7.** Measurement error of the paired-tree distance (ΔD) before and after model correction and spatial accumulation. (**a**) ΔD between trees based on intra- and inter-point cloud data before and after correction using GLM. "×" denotes the mean value and "***" denotes a significant difference according to the one-way analysis of variance ($p < 0.001$). (**b**) Differences in the mean and standard deviation of ΔD as the search radius increased. The black line in the figure is the curve fitted by locally weighted regression.

According to Figure 7b, the mean of ΔD measured with the raw point clouds increased as the search range increased, thereby indicating a significant cumulative spatial effect on ΔD. When two trees were from different point cloud data, the spatial cumulative effect was more significant. When the search radius was less than 100 m, the standard deviation of ΔD had a positive correlation with the search radius. When the search radius was greater than 100 m, the variation in the standard deviation of ΔD decreased. After correction, the mean and standard deviation of ΔD decreased significantly, especially for trees from different point cloud data, and the relationship with the search radius was closer to an S-shaped curve. The mean ΔD values did not increase significantly when the search radius was less than 200 m and greater than 400 m.

## 4. Discussion

### 4.1. Efficiency of Backpack Lidar in Forest Inventory

In this study, we verified the suitability of backpack lidar for forest inventory in a large and spatially complex area. In total, 4.5 machine hours were required to complete lidar scanning of a 10 ha forest sample plot. Ground-based TLS equipment requires leveling and calibration, and the layout must be selected for the control points in forests with undulating terrain containing many obstacles, which are all time-consuming processes. At least 1 h and 10 h are needed when using TLS to complete forest scanning at areas of 30 m × 30 m and 1 ha, respectively [19,32]. The subsequent point cloud registration process also requires a large amount of work. Therefore, few studies have used TLS to collect forest structure data in an area larger than 1 ha. In addition, airborne laser scanning is highly efficient, but it is difficult to obtain forest information below the canopy, such as DBH. It is also difficult to accurately locate single trees in a forest with high canopy connectivity and overlap. Hyyppä et al. [33] used a lightweight, small, unmanned aerial vehicle (UAV) that flew under the canopy carrying lidar equipment to scan a forest plot measuring 30 m × 30 m. The flight required 10–15 min and the accuracy of tree structure measurements was similar to that obtained using TLS. However, they studied a small forest plot and the tree density was low. The stability and safety of UAV flights under the canopy of large, closed forests need to be further investigated. Thus, backpack lidar seems to be the most efficient and convenient remote sensing approach for determining the natural forest inventory in a large area.

### 4.2. Factors Affecting DBH Measurement Accuracy

In the present study, the average DBH measurement error was 4.19 cm, which was higher than that obtained previously in small plantation [8] and natural forest plots [3] using the same equipment. We found that the incompleteness of the tree stem point cloud was the main source of the DBH measurement error; thus, obstacles in the forest prevented trees from being scanned in all directions and limited the accuracy of the cylinder-fitting algorithm. This explains why the accuracy of the DBH measurements obtained by single station TLS was generally lower than that using multi-station TLS in previous studies [10,31]. In addition, as the DBH increased, the lidar estimations gradually changed from greater than the field measured values to less than these values. The lidar measurement error was smallest when the DBH was about 20 cm. When the tree is too small, the stem surface returns have fewer valid reference points; thus the surrounding noise points would comprise a larger percentage in all cloud points used to fit the cylinder, resulting in an over-estimated DBH, as well as a larger error. When the tree is too large, the shape of the stem cross-section tends to become irregular, resulting in a larger geometric deviation between the $DBH_{field}$ measured by a diameter ruler and $DBH_{lidar}$ estimated with cylinder fitting on point clouds. The lidar DBH measurement error decreased initially and then increased as the PCD increased, and similar results were obtained experimentally in the plantation using the same equipment [8]. However, it is unclear why the point intensity, an indicator of tree surface roughness, contributed little to the measurement error. These indicators only explained 32% of the lidar measurement error, but each single independent variable had a clear linear relationship with the marginal response in terms of the measure-

ment error. The correction model reduced the error by 44.3%. Thus, factors not considered in the present study, including the shape characteristics of the stem section, could have influenced the reliability of the DBH values measured using a ruler. In addition, due to the thick herb cover and litter layer in the plot, the heights determined by lidar and field measurements could differ from 1.3 m, which also indirectly affected the accuracy of the DBH measurements.

### 4.3. Factors Influencing Tree Positioning

Compared with the lidar measurements of tree structure indexes, such as DBH, few studies have investigated the tree positioning accuracy [34–37]. In general, previous studies of tree positioning focused on the accuracy of the single tree segmentation algorithm [38–40], but few have investigated the location accuracy of trees detected from point cloud data. Xie et al. [41] determined the spatial point pattern for trees distributed in a forest plot of 1 ha using lidar, and the patterns differed from the results measured in the field. In addition to the accuracy of tree recognition, the tree positioning accuracy is also critical for studying forest pattern, interspecific relationships, and density effect. We found that, due to obstacles blocking the GNSS signal, the accuracy of tree positioning by backpack lidar varied greatly, and the error range was similar to that in previous studies [17,42]. However, it should be noted that the plot scale, forest density, and data acquisition complexity were much greater in the present study compared with previous investigations, and we attributed the positioning error to factors related to machine learning.

The distance and cumulative route length of the trajectory from trees to the starting point (DS) contributed significantly to the positioning error. A fixed GNSS signal was only present at the starting position outside the forest plot, so DS referred to the distance between the equipment and the GNSS signal. In the present study, the positioning error was affected little by the distance within 280 m of the GNSS signal, thereby demonstrating that the equipment could accurately position trees in the forest within 280 m of a single GNSS signal source. However, the error increased significantly when the distance exceeded 280 m. Thus, 280 m seems to be the maximum range with good data accuracy in the absence of a GNSS signal. Beyond this distance, $\Delta P$ increased with increasing DS, but the slope decreased gradually; this may have been due to the limitation of plot area, and the number of trees with larger DS was lower. However, this did not affect the normality of the sample data and the conclusion that backpack lidar indirectly led to the underestimation of the forest density at a large scale.

Thus, when planning the data acquisition paths in a wider area, we should investigate more open areas around the sample plot, such as roads, in order to set multiple starting points around the sample plot to increase the number of GNSS signal sources and avoid moving the equipment an excessive distance from the start, thereby improving the efficiency and saving power. We could also place highly reflective georeferenced objects on a, for example, 1.3 m pole as ground control points throughout the plot; however, the extremely high density of tree branches may block these control points.

The measurement error increased with the route length, and this was the second most important factor related to the positioning error. This error was due to the SLAM algorithm, which is the basis of the point cloud construction method. SLAM matches adjacent frames in the point cloud based on similar features. The registration of a subsequent frame depends on the previous frame, so if one of the frames is offset, the error will propagate during the registration of each subsequent frame [43]. A common solution to this problem involves implementing a closed-loop optimization process by crossing or overlapping the trajectory [21,44]. When a closed loop occurs, the system will compare whether the current frame is similar to the frame at any previous time according to the topology, then attempt to optimize the data collected between the two previous times to minimize the data accumulation error. In the present study, we did not implement a closed loop in the forest plot, and the trajectory only intersected outside the target forest area near the start and end positions. Figure 7 shows that the marginal response of the error relative to the

route length tended to gradually decrease at the end. A forest environment is complex and navigation equipment failures will occur because the GNSS signal may be blocked, so it would be difficult to complete a specific number of closed loops while maintaining the stability of the inertial navigation system in the equipment. In future experiments, a certain number of reference objects will be preset in a complex forest environment to use as references for creating a closed loop.

The environmental factors around trees and the characteristics of trees contributed little to the positioning error, thereby demonstrating that the equipment is suitable for use in a complex forest environment. However, the complexity of the forest environment may affect the variability of the laser incident surface and features that can be referenced by SLAM, which could not be measured directly. The multivariate linear models based on the marginal response relationships explained less of the variation in the measurement error than the random forest model, which indicates that some nonlinear relationships could have affected the positioning accuracy, and they require further investigation.

*4.4. Paired-Tree Distance*

Paired-tree distance is more important than the precise geographical coordinates of trees in forest ecology studies, but the accurate measurement of the former also determines that of the latter. We found that the measurement error for the paired-tree distance using lidar also increased as the plot area increased. Thus, the estimated forest density was often greater than the actual value when backpack lidar was used for large-scale forest inventory. Considering the endurance capacity of the equipment, multiple scans must be performed, and thus the measurement error of paired-tree distance, which further increases the possibility of underestimating of the forest density. After tree position correction, we found that the paired-tree distance measurement error decreased, and the accuracy of forest density measurements decreased indirectly. In most previous studies based on point clouds without geographical coordinates, the iterative closest point (ICP) algorithm was used to register the point cloud data [45]. The ICP algorithm involves a transformation using the least-squares method based on one point cloud, and translates or rotates another point cloud to minimize the sum of squares of the distances between the same objects in two datasets [46]. The ICP algorithm involves a rigid transformation, so the data positioning error is difficult to eliminate. Moreover, the mandatory manual registration based on visual recognition makes it difficult to determine the benchmark, which will also lead to unknown deformations of the point cloud and make the error unpredictable. The geographic coordinate information provided by GNSS greatly simplifies the registration process. The relationship determined between the positioning error and plot scale, as well as the error accumulation identified with the SLAM algorithm in the present study, may facilitate the development of a more refined registration algorithm.

**5. Conclusions**

Backpack lidar integrated with GNSS and the SLAM technique is an effective tool for tree inventory in a large area of closed forest. However, when the GNSS signal is blocked, single tree positioning based on the lidar point cloud can result in remarkable errors, which indirectly lead to the underestimation of the forest density at a large scale. The positioning error had strong correlations with the distance of the scanned tree from the GNSS signal and the cumulative effect of the error caused by the SLAM algorithm, but weak correlations with the environmental complexity and tree properties. Moreover, the properties of trees, such as the DBH, and obstruction by other trees in the forest affected the accuracy of the DBH measurements by lidar. With the influential factors accounted for, the DBH measurement error and offset of the tree positioning in different directions were significantly reduced. The accuracy of measurements of the paired-tree distance at different scales was also optimized. Our results provide a helpful test for the potential of applying backpack lidar for data collection in complex forests, with useful hints for data quality improvement.

**Author Contributions:** Conceptualization, Z.S. and Y.X.; methodology, Y.X. and Z.S.; formal analysis, Y.X.; software, S.P.; investigation, Y.X., T.Y., X.W. and X.C.; resources, J.H., A.W. and L.C.; data curation, Y.X., T.Y., X.W. and X.C.; writing—original draft preparation, Y.X.; writing—review and editing, Z.S.; visualization, Y.X.; funding acquisition, Z.S. All authors have read and agreed to the published version of the manuscript.

**Funding:** This study is sponsored by the Natural Science Foundation of China (No. 41790425).

**Institutional Review Board Statement:** Not applicable.

**Informed Consent Statement:** Not applicable.

**Data Availability Statement:** Not applicable.

**Acknowledgments:** We are grateful for the support from the Zhujiangyuan Provincial Natural Reserve.

**Conflicts of Interest:** The authors declare no conflict of interest. The funders had no role in the design of the study; in the collection, analyses, or interpretation of data; in the writing of the manuscript, or in the decision to publish the results.

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
