# Peer review of "Applying a Portable Backpack Lidar to Measure and Locate Trees in a Nature Forest Plot: Accuracy and Error Analyses"

_remotesensing, doi:10.3390/rs14081806_

Round 1
Reviewer 1 Report
Dear Authors,
This is a nice investigation of the use of a backpack LiDAR in a dense natural forest. Both tree position and tree diameter investigations are worthy contributions to the current body of literature. It seems like a problem was the GPS reception under canopy and the authors had to adjust the data collection strategy. Maybe add some more details and possible solutions that would alleviate this problem.
A few terms were used that I was unfamiliar with. These could have been lost in translation to English. Please see below for details. (Lines 39, 119, 383) Moderate to minor changes required.
Overall a well written and interesting paper. A few suggestions for improvement are found below.
Abstract
No comments. Looks good.
Line 39. What is a “construction investigation” plot?
Line 70. Add a space.
Line 94. Relative distance? I believe the distance between trees is absolute. Or please explain what you mean with ‘relative’.
- Materials and methods.
Line 115. Before -> while
Line 116. a sample plot.
Line 119. What are forest windows? Canopy gaps?
Line 122. Decent map, a little cramped, good use of colors. Maybe add a better photo of the plot. 1c and 1d are small and hard to see.
Line 144. Please cite the software.
Line 146. We split the data acquisition in four parts?
Line 155. Please cite the software.
Line 161-167. Very clear!
Line 170. Again, please explain the use of “relative” distance. Relative to what?
Line 204. How does DBHfield affect the DBH measurement? Does it?
Lines 240-254. This is great!
Line 277. So a more complete point cloud is better for predicting the DBH, am I right?
Line 287. So this method works better with larger trees.
Line 291. Why is the ΔDBH minimal at a PCD of 8 points/cm2? What is it about that density that is optimal?
Line 329. ‘Distance on trajectory’ and ‘Distance to start’ are operational data collection problems, specific to your plot. Glad you addressed them in the discussion. See next comment too.
Line 351. Can you explain these inflection points? What happens figure 6c when DS is 200m (ΔX) or 350m (ΔY)?
And in figure 6b: what happens when the Distance to start is 280m? I.e. what is special about these distances? What causes the slope changes?
Line 383. Maybe ‘inner’ should say ‘intra’. As in: within the same dataset.
- Discussion
Line 434. Again, why is this? (Same as Line 291 comment)
Line 468. Why 300m? Would the addition of a GPS base station at the starting point help? (Maybe use RTK?)
Line 494. Yes! Highly reflective objects (cubes or spheres) on a for example 1.3m pole, which are surveyed or GPS’d (i.e. a good X, Y, and Z) and spread throughout the plot, would help you when tying point clouds to these GCPs!
Line 511. Multiple scans.
Good. No comments.
Author Response
Dear reviewer:
Thank you for your review, which help much for us to improve our revised manuscript..Here are my answers to each of your comments.
Please see the attachment

Reviewer 2 Report
The paper considers a tree map building process using backpack LiDAR scanning. The area is large, as large as another recent study (8-10 ha). The scanning path resembles the work patterns occurring in the modern forestry. Technology and methodology does not yet reach to as accurate DBH measurement as practise requires, but this is the level of research on producing large area tree maps in general.
Two aspects are excellent:
- integrity of PC, which is a nice geometrical measure of the scan quality
- Fig. 3, which clearly presents a set of the contributors to the DBH error
I suggest a further study with the acquired data where you try to add consistency to the overlapping zones of separate scans by using possible systematic features contained within the match error. This would be a machine learning problem similar to separating the signal and error.
Author Response
Dear reviewer:
Thank you for your recognition and support of this study. We are willing to adopt your suggestions in future studies, conduct further research on the acquired data.

Reviewer 3 Report
The aurors of the text of the paper deal with the methods of determining the exact position of the tree and measuring its structural parameters. These parameters are the basis of forest mapping and inventory, which are important for forest management. They propose the use of a portable backpack lidar to calculate biomass and analyze forest community dynamics
which integrates simultaneous localization and mapping (SLAM) techniques. With a Global Navigation Satellite System receiver, this solution then has greater flexibility. However, LIDAR has never been used to inventory trees other than by ground-based laser scanning to measure and map forest structure over a large area of high-density vegetation.
In this study, the authors used LiBackpack DG50 lidar to acquire point cluster data over a 10ha area of subtropical forest and then used this data to quantify errors and associated factors in diameter at breast height (DBH) measurements for over 1900 individual trees. A mean error of 4.19 cm in DBH measurements obtained by lidar was evaluated compared to manual field measurements. The average tree positioning error was 4.64 m and was significantly affected by the distance and length of the route from the measured tree object to the measured object
initial position of the data collection, while it had little effect on it.
The authors proposed an error correction method. After correction, the DBH measurement error was reduced by 31.3%, the tree location error was reduced by 44.3%, and the tree position measurement error was reduced by 4.5%, and the relative distance error was reduced by 56.3%.
The work is technically interesting, it is based on the principles of scientific posupt and the theoretical assumptions were demonstrated in experiments with error evaluation.
Author Response
Dear reviewer:
Thank you for your recognition and support of our study. We would go on to explore and improve the potential of the lidar-based measuring technique in forestry inventary and related experimental studies.

Reviewer 4 Report
Summary:
Thank you for the opportunity to review this great manuscript. The manuscript titled “Applying a portable backpack lidar to measure and locate trees in a nature forest plot: accuracy and error analyses” assess the performance of the LiBack- 20 pack DG50 backpack lidar system to measure tree DBH and tree locations in a densely populated natural forest area. The study brings a great point of using backpack lidar systems in place of Terrestrial laser scanning system that requires both high investment in money and time. Study analyzed around 1900 individual tree locations an their DBH measured using total stationZ TS-420R and a DBH tape to evaluate the performance of the backpack lidar system derived point clouds. Initial results indicate a 4.19 cm in DBH measurements error and 4.64 m tree positioning error compared to field estimates. However, the errors were reduced by 31.3% for DBH and by 44.3% for tree positioning after applying correction factors estimated using a model developed using potential influencing factors such as integrity of point cloud, field measured DBH, and point density. Overall, the manuscript has been greatly designed and written. The results and discussion have been explained very clearly. Other than the minor edits required below the manuscript can be accepted in the correct form.
Broader impact:
The study brings a great point of using backpack lidar systems in place of Terrestrial laser scanning system that requires both high investment in money and time. The manuscript provides a significant contribution by assessing a backpack lidar system to estimate tree positioning and structure metrics in a densely populated forest area. Overall, the manuscript is well written. Please revise following minor errors in the final version.
L118 : 8600 trees/hm2 is this 8600 trees/ha or 8600 trees/m2 ?
L 118: canopy density was over 90% compared to what or how did you estimate this?
L114: reference for LiFuser BP? Please add all the references for new software used to collect and process your data including statistical estimates.
L149: space between section number and section title (e.g. 2.4.1Preprocessing)
L155: reference for LiDAR 360 V5.0 software?
L168: Figure 2a?
L188: (Fig. 2c) ?
L236: Can you please point out which R packages used instead R version?
L271: regressionto (space is missing between two words)
L305: Subscript field and real in DBHfield as DBHreal to be consistent
Figure 4: need description about the blue line.
Figure 5: in figure 5b, are these different datasets in the figure 1? If so, please add it to the caption.
L402: Space between forest and inventory
L465-L475: How about using ground control points? Can we increase the point cloud accuracy? Please add few words on this to this section.
Author Response
Dear reviewer:
Thank you for your review, which help much for us to improve our revised manuscript. Here are my answers to each of your comments.
Please see the attachment
